 # Liger: Linearizing Large Language Models to Gated Recurrent Structures

Disen Lan [1 2 *]  Weigao Sun [1 ✉]  Jiaxi Hu [3]  Jusen Du [1 4 *]  Yu Cheng [5 ✉]

## Abstract

Transformers with linear recurrent modeling offer linear-time training and constant-memory inference. Despite their demonstrated efficiency and performance, pretraining such non-standard architectures from scratch remains costly and risky. The linearization of large language models (LLMs) transforms pretrained standard models into linear recurrent structures, enabling more efficient deployment. However, current linearization methods typically introduce additional feature map modules that require extensive fine-tuning and overlook the gating mechanisms used in state-of-the-art linear recurrent models. To address these issues, this paper presents **Liger**, short for **Li**nearizing LLMs to **g**at**e**d **r**ecurrent structures. Liger is a novel approach for converting pretrained LLMs into gated linear recurrent models without adding extra parameters. It repurposes the pretrained key matrix weights to construct diverse gating mechanisms, facilitating the formation of various gated recurrent structures while avoiding the need to train additional components from scratch. Using lightweight fine-tuning with Low-Rank Adaptation (LoRA), Liger restores the performance of the linearized gated recurrent models to match that of the original LLMs. Additionally, we introduce Liger Attention, an intra-layer hybrid attention mechanism, which significantly recovers 93% of the Transformer-based LLM performance at 0.02% pre-training tokens during the linearization process, achieving competitive results across multiple benchmarks, as validated on models ranging from 1B to 8B parameters.

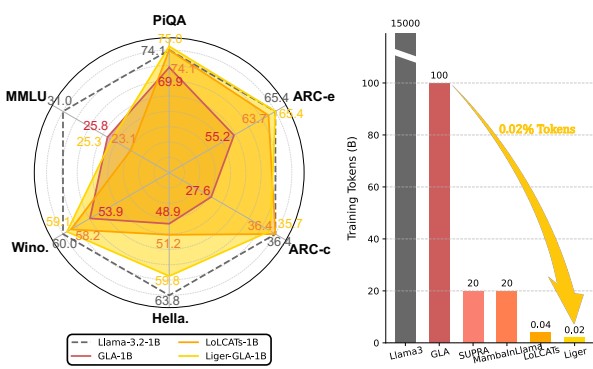

*Figure 1.* **Liger Performance and Efficiency.** Our proposed Liger recovers nearly 93% performance of Llama-3.2-1B and outperforms pretrained gated recurrent models at only 0.02% of the pre-training tokens cost.

## 1. Introduction

Large language models (LLMs) have demonstrated exceptional performance across various natural language processing tasks (Chintala, 2023; Team, 2023; Zhu et al., 2024; Qu et al., 2024). However, the Transformer-based architecture (Vaswani et al., 2017) used in modern LLMs, with its reliance on softmax attention, suffers from quadratic computational complexity. This inefficiency results in significant speed and memory challenges, particularly during pretraining on long sequences. During inference, the Key-Value (KV) cache (Kwon et al., 2023) grows linearly with the input sequence length, leading to reduced inference speed and high memory usage, which severely limits the capability of these models for handling long-sequence tasks (Sun et al., 2024a). In contrast, models based on linear recurrent modeling (Katharopoulos et al., 2020; Yang et al., 2023; Qin et al., 2024b; Sun et al., 2025; Du et al., 2025) provide linear-time training and constant-memory inference, offering substantial efficiency benefits and positioning themselves as promising candidates for the next generation of foundational architectures (MiniMax et al., 2025).

While pretraining LLMs using architectures based on linear

[1]Shanghai AI Laboratory [2]South China University of Technology [3]The Hong Kong University of Science and Technology (Guangzhou) [4]Nanjing University [5]The Chinese University of Hong Kong. * Interns at Shanghai AI Laboratory. ✉ Corresponding Authors: Weigao Sun <sunweigao@outlook.com>, Yu Cheng <chengyu@cse.cuhk.edu.hk>. Weigao Sun is the Project Lead.

*Proceedings of the 42nd International Conference on Machine Learning*, Vancouver, Canada. PMLR 267, 2025. Copyright 2025 by the author(s).

The source code is available at https://github.com/OpenSparseLLMs/Linearization and the models are available at https://huggingface.co/collections/linear-moe-hub.

recurrent modeling reduces costs due to their linear training complexity, the high expenses of pretraining from scratch associated with large model sizes and datasets still remain a major obstacle to their adoption and practical use. This challenge has hindered the advancement of linear recurrent models. Linearizing pretrained LLMs like SUPRA (Mercat et al., 2024), MambaInLlama (Wang et al., 2024) and LoL-CATs (Zhang et al., 2024a), as an emerging new direction, allows the transfer of weights from an existing pretrained model to one with linear recurrent modeling architectures at a small fraction of the original pretraining cost. The linearization approach is a promising post-training technique to enable efficient pretrained model deployment while preserving their performance. Gating mechanisms (Qin et al., 2024a; Sun et al., 2023) play a crucial role in linear recurrent models by controlling memory retention and forgetting, with their effectiveness widely demonstrated in such architectures. However, incorporating gate modules as additional components requires both transferring weights from pre-trained LLMs and training these gating modules from scratch. This process not only increases the cost of linearization but also creates a larger architectural divergence from Transformer-based LLMs. This divergence may hinder the effective approximation of softmax attention, limiting the performance of gated linear recurrent models (Zhang et al., 2024d). Moreover, existing linearization methods often overlook the detailed design considerations of gated linear models, and the newly added modules fail to leverage the pre-trained weights of LLMs, further reducing the efficiency of linearization.

In this paper, we present **Liger**, which stands for **Li**nearizing large language models to **g**ated **r**ecurrent structures, a novel approach for linearizing LLMs. Liger repurposes the weights from pre-trained Transformer-based LLMs and introduces a novel method for constructing crucial gating mechanisms in gated recurrent structures using the key projection. This approach avoids the complex attention transfer process found in existing linearization methods. After transforming the weights and constructing the gating mechanisms, Liger requires only lightweight fine-tuning of the linearized gated recurrent model parameters through LoRA autoregressive training. By introducing Liger Attention, this efficient process restores further improves the model's performance with minimal linearization cost, achieving competitive results across a range of language modeling and understanding benchmarks while benefiting from the linear-time inference efficiency of the recurrent architecture.

Our contributions can be summarized as follows:

- We introduce **Liger**, a novel method for adapting pre-trained Transformer-based LLMs into gated recurrent structures. This approach efficiently repurposes redundant weights from pre-trained models to construct

gating modules without introducing additional parameters, obtaining gated recurrent LLMs with the benefits of constant-memory inference.

- We propose **Liger Attention**, an intra-layer hybrid attention mechanism that combines sliding window softmax attention with linear recurrent modeling. This simple yet effective design retains the essential softmax non-linearity, accelerating the linearization process while maintaining the capabilities of pre-trained LLMs and ensuring linear-time inference efficiency.

- We apply Liger to linearize the latest Llama-3 series, ranging from 1B to 8B parameters. Experimental results show that Liger outperforms existing linearization methods (like SUPRA (Mercat et al., 2024), MambaIn-Llama (Wang et al., 2024) and LoLCATs (Zhang et al., 2024a)), in terms of both efficiency and its ability to preserve the original performance of pre-trained LLMs.

## 2. Preliminary

**Transformer with Softmax Attention.** Given the input sequence $\mathbf{X} = \{\boldsymbol{x_1}, \boldsymbol{x_2}, \ldots, \boldsymbol{x_T}\} \in \mathbb{R}^{T \times D}$, with sequence length $T$ and dimension $D$, vanilla transformer (Vaswani et al., 2017) adopts standard softmax attention:

$$\mathbf{Q}, \mathbf{K}, \mathbf{V} = \mathbf{X}\mathbf{W_Q}, \mathbf{X}\mathbf{W_K}, \mathbf{X}\mathbf{W_V}$$
$$\mathbf{O} = \text{Softmax}((\frac{\mathbf{Q}\mathbf{K}^\top}{\sqrt{D}}) \odot \mathbf{M})\mathbf{V} \quad (1)$$

where $\mathbf{W_Q}, \mathbf{W_K}, \mathbf{W_V} \in \mathbb{R}^{D \times D}$ are learnable parameters for input sequence $\mathbf{X}$ projection and $\mathbf{M} \in \mathbb{R}^{T \times T}$ is a mask matrix for causal modeling by preventing future information leakage in autoregressive generation task. The above *parallel form* of softmax attention in Eq.1 is applied for efficient training and can be rewritten in the following *recurrent form* during inference stage:

$$\boldsymbol{q}_t, \boldsymbol{k}_t, \boldsymbol{v}_t = \boldsymbol{x}_t\mathbf{W_Q}, \boldsymbol{x}_t\mathbf{W_K}, \boldsymbol{x}_t\mathbf{W_V}$$
$$\boldsymbol{o}_t = \frac{\sum_{i=1}^{t} \exp(\boldsymbol{q}_t\boldsymbol{k}_i^\top / \sqrt{D})\boldsymbol{v}_i}{\sum_{i=1}^{t} \exp(\boldsymbol{q}_i\boldsymbol{k}_i^\top / \sqrt{D})} \quad (2)$$

The standard softmax attention is highly reliant on the growing KV Cache (Chou et al., 2024b; Wang et al., 2024) to recall the history "memory" for sequence modeling, which results in quadratic complexity and costly memory requirements especially in long context setting.

**Linear Attention.** Linear transformer (Katharopoulos et al., 2020; Qin et al., 2023) approximates softmax self-attention as the dot product of the kernel feature mapping and utilizes associative property of matrix products to calculate the self-attention weights, achieving efficient linear-time sequence

modeling and constant memory consumption. Concretely, the linear attention can be formulated as follows:

$$o_t = \frac{\sum_{i=1}^{t} \phi(\boldsymbol{q_t})\phi(\boldsymbol{k_i})^\top \boldsymbol{v_i}}{\sum_{i=1}^{t} \phi(\boldsymbol{q_t})\phi(\boldsymbol{k_i})^\top}$$
$$= \frac{\phi(\boldsymbol{q_t})\sum_{i=1}^{t} \phi(\boldsymbol{k_i})^\top \boldsymbol{v_i}}{\phi(\boldsymbol{q_t})\sum_{i=1}^{t} \phi(\boldsymbol{k_i})^\top} \tag{3}$$

Let $\mathbf{S}_t = \sum_{i=1}^{t} \phi(\boldsymbol{k_i})^\top v_i$ and $\boldsymbol{z_t} = \sum_{i=1}^{t} \phi(\boldsymbol{k_i})^\top$, the above formulation in Eq. 3 can be rewritten in the *recurrent form* as an RNN (Katharopoulos et al., 2020):

$$\begin{cases} \mathbf{S}_t = \mathbf{S}_{t-1} + \phi(\boldsymbol{k_t})^\top \boldsymbol{v_t}, \\ \boldsymbol{z_t} = \boldsymbol{z_{t-1}} + \phi(\boldsymbol{k_t})^\top, \end{cases} \quad o_t = \frac{\phi(\boldsymbol{q_t})\mathbf{S}_t}{\phi(\boldsymbol{q_t})\boldsymbol{z_t}} \tag{4}$$

Although linear attention with causal mask matrix cannot use matrix associativity to reduce the *parallel form* training complexity from quadratic to linear, its *chunk-wise parallel form* allows hardware-efficient sub-quadratic and partially parallel training (Yang et al., 2023; Sun et al., 2024a; 2025; Qin et al., 2024a).

**Gating Mechanism.** While linear attention (or linear recurrent structures) are widely recognized for their linear-time computational efficiency, they have historically exhibited a notable performance gap compared to standard softmax attention. To address this limitation, recent advances in linear recurrent models have incorporated gating mechanism, which is a critical architectural component enabling dynamic, context-aware information retention through input and forget gates. This mechanism allows models to selectively preserve or discard historical information, substantially enhancing their expressiveness through constant memorization capacity. The integration of gating mechanism has consequently become a prevalent design paradigm in state-of-the-art linear attention variants (Yang et al., 2023; Peng et al., 2023; Qin et al., 2024c). A representative implementation, Gated Linear Attention (GLA) (Yang et al., 2023), demonstrates this principle through its mathematical formulation:

$$\mathbf{S}_t = \mathbf{G}_t \odot \mathbf{S}_{t-1} + \boldsymbol{k}_t^\top \boldsymbol{v}_t \tag{5}$$

At its core, Gated Linear Attention (GLA) fundamentally augments conventional linear attention through the integration of a gating mechanism. This modification serves as a generalized framework that can be systematically extended to diverse linear recurrent architectures by reparameterizing the gating term $\mathbf{G}_t$. Specifically, $\mathbf{G}_t$ governs the temporal decay dynamics, enabling broad and flexible adaptation across variant gated linear recurrent structures.

## 3. Methodology

In this section, we will introduce our proposed Liger for linearizing large language models to gated recurrent structures. We also design a simple yet effective hybrid attention form, namely Liger Attention, and build the Liger architecture based on it, including intra- and inter-layer hybrid architectures.

### 3.1. Gate Construction by Key Projection

The parameter space of large language models (LLMs) exhibits intrinsic structural redundancy (Yu et al., 2024; Aghajanyan et al., 2020), a phenomenon attributed to the over-parameterization inherent in deep neural architectures. This redundancy motivates a principled approach to reformulating LLMs as gated linear recurrent architectures: rather than introducing new parameters, we strategically repurpose subsets of pre-trained LLM weights to serve dual roles as gating modules.

Building on the design principle of gated linear recurrent structures for optimal softmax attention approximation, we propose reallocating the key projection matrix $\mathbf{W_K}$ as dual roles to concurrently perform its canonical linear transformation and gating mechanism. Formally, the gating mechanism is derived via a transformation of $\mathbf{G}_t = f(\boldsymbol{k}_t) = f(\boldsymbol{x}_t \mathbf{W}_K)$, where $f(\cdot)$ operates on the projected key embeddings. This parameter-sharing paradigm ensures compatibility with pre-trained weights while eliminating the need for auxiliary trainable gating parameters, thereby preserving computational and memory efficiency.

In practical implementations, gating mechanisms can be instantiated through diverse transformation strategies. Our approach employs a parameter-free $\text{Pooling}(\cdot)$ operation to derive gate values, circumventing the need for additional trainable parameters. This design preserves compatibility with pre-trained LLM weights, enabling direct reuse of existing parameters for gate construction without architectural modification. Empirical evaluations demonstrate that this parameter-efficient strategy achieves competitive performance compared to conventional trainable gating projections (e.g., linear or nonlinear parametric layers), while maintaining computational efficiency and reducing optimization complexity.

### 3.2. Liger: Linearizing LLMs to Gated Recurrent Structures

Prior methodologies for linearizing transformer-based large language models (LLMs) typically rely on auxiliary components, such as feature mapping layers, to approximate softmax attention mechanisms (Zhang et al., 2024a). Notably, LoLCATs (Zhang et al., 2024a) propose a two-stage fine-tuning paradigm to mitigate this limitation: *First stage*:

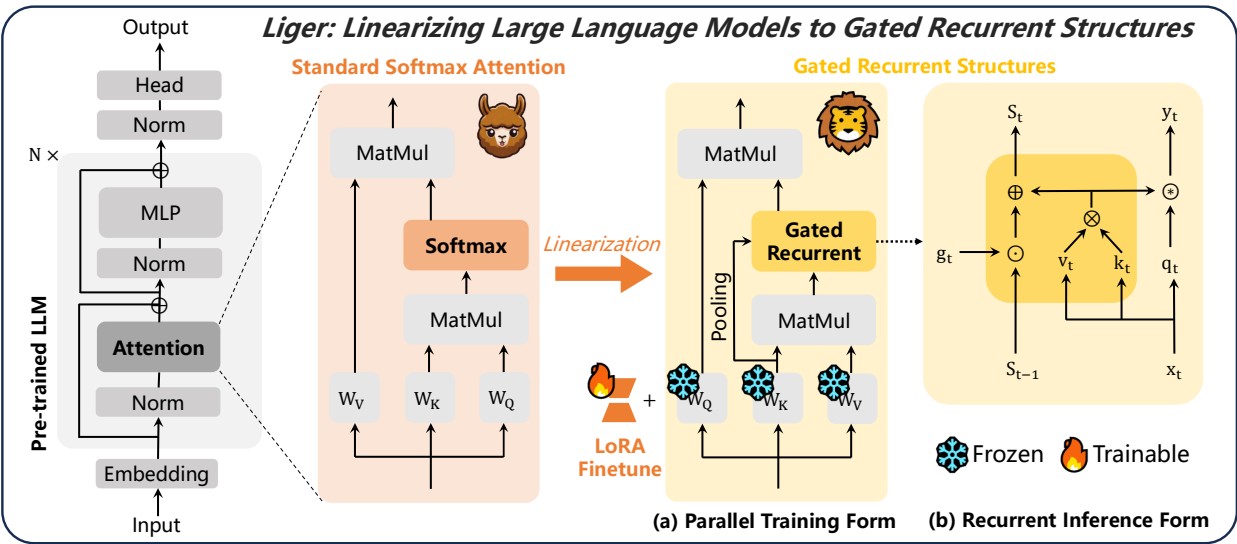

*Figure 2.* **Overall Framework of Liger.** We linearize the Transformer-based large language model (LLM) architecture into a gated linear recurrent model by 1) Replacing *Softmax Attention* with a *Gated Recurrent Memory* module, and 2) Employing *LoRA* to fine-tune the Liger architecture while frozen most original weight parameters. The Liger architecture enables efficient chunk-wise parallel training also enjoying cheap linear recurrent inference.

| Model | Gate Parameterization | Pooling for Gate Construction |
|---|---|---|
| Gated Linear Attention (Yang et al., 2023) | $\mathbf{G}_t = \boldsymbol{\alpha}_t^\top \mathbf{1}$ | $\boldsymbol{\alpha}_t = \sigma(\text{Pooling}(\boldsymbol{k}_t))$ |
| Mamba2 (Dao & Gu, 2024) | $\mathbf{G}_t = \alpha_t \mathbf{1}^\top \mathbf{1}$ | $\boldsymbol{\alpha}_t = \exp(-\text{softplus}(\text{Pooling}(\boldsymbol{k}_t)))$ |
| mLSTM (Beck et al., 2024) | $\mathbf{G}_t = \alpha_t \mathbf{1}^\top \mathbf{1}$ | $\boldsymbol{\alpha}_t = \sigma(\text{Pooling}(\boldsymbol{k}_t))$ |
| Gated Retention (Sun et al., 2024b) | $\mathbf{G}_t = \alpha_t \mathbf{1}^\top \mathbf{1}$ | $\boldsymbol{\alpha}_t = \sigma(\text{Pooling}(\boldsymbol{k}_t))$ |
| HGRN2 (Qin et al., 2024c) | $\mathbf{G}_t = \boldsymbol{\alpha}_t^\top \mathbf{1}$ | $\boldsymbol{\alpha}_t = \gamma + (1-\gamma)\sigma(\text{Pooling}(\boldsymbol{k}_t))$ |
| RWKV6 (Peng et al., 2024) | $\mathbf{G}_t = \boldsymbol{\alpha}_t^\top \mathbf{1}$ | $\boldsymbol{\alpha}_t = \exp(-\exp(\text{Pooling}(\boldsymbol{k}_t)))$ |
| Gated Slot Attention (Zhang et al., 2024c) | $\mathbf{G}_t = \boldsymbol{\alpha}_t^\top \mathbf{1}$ | $\boldsymbol{\alpha}_t = \sigma(\text{Pooling}(\boldsymbol{k}_t))$ |

*Table 1.* **Gated Linear Recurrent Structures with Variations of Gate $\mathbf{G}_t$ Parameterization.** Gating mechanism can be constructed through pooling to reuse the key projection of pre-trained LLM.

Attention Transfer phase trains newly introduced modules (e.g., kernel approximations) while freezing pre-existing parameters, followed by the *Second stage*: employing Low-Rank Adaptation (LoRA) to fine-tune attention layers. However, previous linearization approaches including LoLCATs incurs two critical constraints:

❶ *Architectural Overhead*: The dependency on supplementary feature mapping and gating modules to replicate softmax attention outputs precludes direct reuse of pre-trained LLM parameters, necessitating non-trivial architectural modifications.

❷ *Optimization Fragility*: The sequential training paradigm introduces brittleness, as end-to-end fine-tuning is infeasible due to the interdependency between the frozen base model and the auxiliary components.

These limitations hinder extensibility to modern linear recurrent architectures incorporating gated mechanisms, which require seamless integration with pre-trained weights and end-to-end trainability.

To advance the linearization of pre-trained large language models (LLMs) into gated recurrent neural architectures, we propose a parameter-efficient strategy that employs the canonical Softmax operation for feature mapping. Unlike existing approaches that rely on trainable modules such as T2R (Mercat et al., 2024) or Hedgehog (Zhang et al., 2024b) to approximate attention dynamics, our method utilizes Softmax to inherently normalize query and key representations. This normalization ensures bounded magnitude in the query-key product space, a critical property for faithfully replicating the numerical stability of conventional softmax attention within linearized frameworks.

By eschewing auxiliary trainable components, our design eliminates architectural dependencies on attention transfer mechanisms. This divergence from sequential training paradigms (e.g., frozen base models with incremental module updates) enables fully end-to-end fine-tuning without

compromising compatibility with pre-trained LLM weights. The resultant gated recurrent architecture, formulated as:

$$\boldsymbol{q}_t = \boldsymbol{x}_t \mathbf{W_q}, \boldsymbol{k}_t = \boldsymbol{x}_t \mathbf{W_k}, \boldsymbol{v}_t = \boldsymbol{x}_t \mathbf{W_v}$$
$$\mathbf{G}_t = \mathrm{Pooling}(\boldsymbol{k}_t) \tag{6}$$

With the generated gate $\mathbf{G}_t$, the recurrent update rule and the followed output computation will be:

$$\mathbf{S}_t = \mathbf{G}_t \odot \mathbf{S}_{t-1} + \phi(\boldsymbol{k}_t^\top)\boldsymbol{v}_t$$
$$\boldsymbol{o}_t = \phi(\boldsymbol{q}_t)\mathbf{S}_t \tag{7}$$

Here all the trainable parameters $\mathbf{W_Q}, \mathbf{W_K}, \mathbf{W_V}$ are inherited from the pre-trained LLM. This method of gating mechanism construction can be extended to various gated linear recurrent structures, as shown in Table 1. Prior linear recurrent models employ learnable feature mapping functions (denoted as $\phi(\cdot)$) to compute similarity representations between query ($\boldsymbol{q}_t$) and key ($\boldsymbol{k}_t$) vectors, which introduce superfluous trainable parameters while generating output distributions that deviate from the canonical $\mathrm{Softmax}$ attention distribution inherent in pre-trained LLM. Such distributional discrepancies consequently degrade the efficacy of linearization by impairing compatibility with the original attention mechanisms. We found that feature mapping $\phi(\cdot)$ can be effectively approximated via a simple normalization operation ($\mathrm{Softmax}(\cdot)$ in our implementation), thereby eliminating the requirement for computationally intensive attention transfer or distillation procedures while maintaining fidelity to the target attention distribution. Eq. 7 preserves the expressivity of softmax attention while inheriting the computational efficiency of linear recurrent models. This unification of architectural simplicity and functional fidelity addresses key limitations in prior linearization methods.

Following the initialization of the model with pre-trained LLM weights and its architectural reconfiguration into a gated linear recurrent framework, we employ *next-token prediction* as the fine-tuning objective to recover performance in the transformed architecture. Formally, we parameterize the adapted weights as $\Theta = \Theta_0 + \Delta\Theta$, where $\Delta\Theta$ denotes the incremental adjustments required to align the original transformer-based parameters with the gated linear recurrent structure. The optimization objective minimizes the cross-entropy loss $\mathcal{L}$ for autoregressive prediction over input sequences $x_{1:t-1}$:

$$\mathcal{L} = -\sum \log P_\Theta(\boldsymbol{x}_t | \boldsymbol{x}_{1:t-1}) \tag{8}$$

This approach circumvents the need for auxiliary training stages (e.g., attention transfer) by directly optimizing the gated linear recurrent architecture end-to-end, thereby preserving the computational efficiency and parameter efficiency inherent to the original LLM.

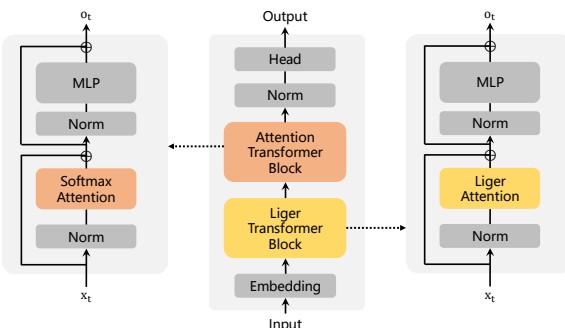

Figure 3. **Liger Hybrid Architecture.** Liger adopts intra-hybrid Liger Attention and inter-hybrid model architecture by stacking a layer of standard attention Transformer blocks every a few (e.g. 7) layers of Liger Transformer blocks.

we apply Low-Rank Adaptation (LoRA) (Hu et al., 2021) specifically to the linear recurrent layers of large language models (LLMs), focusing on the fine-tuning of the weight matrices $\mathbf{W_Q}, \mathbf{W_K}, \mathbf{W_V}$. Instead of training all model parameters, LoRA decomposes the adaptation term $\Delta\Theta$ into two low-rank matrices $\mathbf{B}$ and $\mathbf{A}$, such that $\Delta\Theta = \mathbf{BA}$, where $\mathbf{B} \in \mathbb{R}^{D \times r}$, $\mathbf{A} \in \mathbb{R}^{r \times D}$, and $r \ll D$ with $r$ typically set to a small value, such as 8. Our empirical results demonstrate that LoRA consistently outperforms full-rank fine-tuning, offering a more efficient and effective approach for LLM linearization.

### 3.3. Liger Attention: A Hybrid of Sliding Window Attention and Gated Recurrent Modeling

Building upon previous works that combine softmax attention with linear attention (Katharopoulos et al., 2020; Arora et al., 2024; MiniMax et al., 2025), we propose an intra-layer hybrid attention mechanism, termed **Liger Attention**. This method integrates a hybrid form of Gated Recurrent Modeling (GRM) and Sliding Window Attention (SWA) (Beltagy et al., 2020) with narrow softmax attention window size, by blending their outputs in a weighted manner. Specifically, the formulation is given by:

$$\boldsymbol{o}_t = \mathrm{LigerAttn}(\boldsymbol{q}_t, \boldsymbol{k}_t, \boldsymbol{v}_t)$$
$$= \alpha\, \mathrm{GRM}(\boldsymbol{q}_t, \boldsymbol{k}_t, \boldsymbol{v}_t) + \beta\, \mathrm{SWA}(\boldsymbol{q}_t, \boldsymbol{k}_t, \boldsymbol{v}_t) \tag{9}$$

where GRM denotes Gated Recurrent Modeling (and its variants) in Eq. 3 and SWA refers to Sliding Window Attention, which is a variant of softmax attention as expressed in Eq. 1 formulated in Eq. 10. The parameters $\alpha$ and $\beta$ control the relative contributions of each attention mechanism. We found that setting the sum of the two parameters $\alpha$ and $\beta$ to 1 is particularly critical for linearization (simply sets to 0.5 each in our implementation), which can better approximate the original attention output distribution.

| Model | Training Tokens (B) | PiQA | ARC-e | ARC-c | Hella. | Wino. | MMLU | Avg. | Avg. |
|---|---|---|---|---|---|---|---|---|---|
| | | acc ↑ | acc ↑ | acc_norm ↑ | acc_norm ↑ | acc ↑ | acc (5-shot) ↑ | ↑ | (no MMLU) ↑ |
| Mistral-7B | 8000 | 80.6 | 80.7 | 53.9 | 81.1 | 74.3 | 62.6 | 72.2 | 74.1 |
| SUPRA-Mistral-7B | 100 | 80.4 | 75.9 | 45.8 | 77.1 | 70.3 | 34.2 | 64.0 | 69.9 |
| LoLCATs-Mistral-7B Attn. Trf. | 0.02 | 79.8 | 79.3 | 51.7 | 48.3 | 74.2 | 23.0 | 59.4 | 66.7 |
| LoLCATs-Mistral-7B LoRA | 0.02 | 77.3 | 74.9 | 45.1 | 40.9 | 67.9 | 23.0 | 54.8 | 61.2 |
| LoLCATs-Mistral-7B | 0.04 | 79.7 | 78.4 | 47.4 | 58.4 | 71.0 | 23.7 | 59.8 | 67.0 |
| Liger-GLA-Mistral-7B (Ours) | 0.02 | 80.1 | 78.7 | 49.3 | 76.3 | 70.1 | 36.3 | 65.1 | 70.9 |
| Llama-3-8B | 15000 | 79.4 | 80.1 | 53.2 | 79.2 | 72.9 | 65.3 | 71.7 | 73.0 |
| SUPRA-Llama-3-8B | 20 | 78.9 | 75.1 | 46.5 | 71.7 | 65.8 | 40.9 | 63.2 | 67.6 |
| Mamba2-Llama-3-8B | 20 | 76.8 | 74.1 | 48.0 | 70.8 | 58.6 | 43.2 | 61.9 | 65.6 |
| Mamba2-Llama-3-8B 50% Attn. | 20 | 81.5 | 78.8 | 58.2 | 79.5 | 71.5 | 56.7 | 71.0 | 73.9 |
| LoLCATs-Llama-3-8B Attn. Trf. | 0.02 | 78.4 | 79.3 | 51.9 | 51.6 | 73.4 | 23.5 | 59.7 | 66.9 |
| LoLCATs-Llama-3-8B LoRA | 0.02 | 72.4 | 72.6 | 44.3 | 34.6 | 68.0 | 23.0 | 52.5 | 58.4 |
| LoLCATs-Llama-3-8B | 0.04 | 80.1 | 80.4 | 53.5 | 63.4 | 72.9 | 42.1 | 65.4 | 70.0 |
| Liger-GLA-Llama-3-8B (Ours) | 0.02 | 80.3 | 81.1 | 52.5 | 76.3 | 72.0 | 43.4 | 67.6 | 72.4 |

*Table 2.* **Linearized LLMs Comparison.** Liger outperforms other linearization method on language modeling and understanding tasks with less training tokens across Mistral-7B and Llama-3-8B LLM architectures.

$$\hat{o}_t = \text{SWA}(q_t, k_t, v_t)$$
$$= \frac{\sum_{i=t-w+1}^{t} \exp(q_t k_i^\top / \sqrt{D}) v_i}{\sum_{i=t-w+1}^{t} \exp(q_t k_i^\top / \sqrt{D})} \quad (10)$$

where $w$ denotes the window size (set to 64 in our default implementation) for limiting the length of the lookback window for the input token. Liger Attention demonstrates strong performance in sequence modeling tasks while maintaining efficient linear complexity of $\mathcal{O}(TWD + TD^2)$.

### 3.4. Liger Architecture and Its Hybrid

The overall architecture of our proposed Liger is presented in Fig. 2. Following the popular LLM architecture like Llama (Dubey et al., 2024), we retain the Pre-Norm layers and MLP layers with residual connection (He et al., 2016), only change the softmax attention layers with Liger attention without introduction of any new trainable modules like feature mapping. For the each Liger blocks including time mixing layer and token mixing layer, the forward process can be formulated as:

$$\mathbf{H} = \text{LigerAttn}(\text{Norm}(\mathbf{X})) + \mathbf{X}$$
$$\mathbf{O} = \text{MLP}(\text{Norm}(\mathbf{H})) + \mathbf{H} \quad (11)$$

As presented in Fig. 3, we also attempt to add one softmax attention block after stacking a number of Liger (or gated linear recurrent) blocks to construct layer-wise hybrid model architecture.

## 4. Experiments

In this section, we conduct extensive experiments to answer the following research questions ($\mathcal{RQ}$):

$\mathcal{RQ}$**1:** Can Liger linearize the pre-trained LLMs and recover performance more effectively compared with other linearization methods?

$\mathcal{RQ}$**2:** Can Liger serve as a universal and scalable linearization method for different LLM architectures?

$\mathcal{RQ}$**3:** Does Liger genuinely achieves linear/subquadratic time complexity and constant memory inference?

$\mathcal{RQ}$**4:** How effective is Liger to its key components?

### 4.1. Experimental Setups

**Models and Datasets.** We select two popular LLM architectures: Mistral-7B (Jiang et al., 2023) and Llama-3-8B (Dubey et al., 2024) as base model for linearization. We opt for GLA (Yang et al., 2023), a general gated linear recurrent model structure, as the basis of the Liger and its hybrid architecture for linearization. We use 50,000 high quality instruction samples of cleaned Alpaca dataset (Taori et al., 2023) during linearization process to improve instruction-following ability and recover LLM performance in language modeling tasks.

**Implementation Configurations and Details.** All experiments are implemented in PyTorch and conducted on single NVIDIA A800 80GB GPU. We opt for AdamW optimizer with a learining rate of $1e^{-3}$. By default, the LoRA rank is set to 8 and alpha is set to 8. The finetuning epochs is 2, which means we only use 100,000 cleaned Alpaca instruction samples (around 0.02B tokens) for gate reccurrent model linearization. We pad the input sequence to 1024 tokens with mini batch size of 1, and set the global batch size to 8 by gradient accumulaltion, following the settings in LoLCATs (Zhang et al., 2024a).

| Model | Training Tokens (B) | PiQA | ARC-e | ARC-c | Hella. | Wino. | MMLU | Avg. | Avg. |
|---|---|---|---|---|---|---|---|---|---|
| | | acc ↑ | acc↑ | acc_norm ↑ | acc_norm ↑ | acc ↑ | acc (5-shot) ↑ | ↑ | (no MMLU) ↑ |
| **(Transformer)** | | | | | | | | | |
| Mistral-7B | 8000 | 80.6 | 80.7 | 53.9 | 81.1 | 74.3 | 62.6 | 72.2 | 74.1 |
| Llama-3-8B | 15000 | 79.4 | 80.1 | 53.2 | 79.2 | 72.9 | 65.3 | 71.7 | 73.0 |
| **(Linear/Subquadratic)** | | | | | | | | | |
| Mamba-7B | 1200 | 81.0 | 77.5 | 46.7 | 77.9 | 71.8 | 33.3 | 64.7 | 71.0 |
| RWKV-6-World-7B | 1420 | 78.7 | 76.8 | 46.3 | 75.1 | 70.0 | - | 69.4 | 69.4 |
| TransNormerLLM-7B | 1400 | 80.1 | 75.4 | 44.4 | 75.2 | 66.1 | 43.1 | 64.1 | 68.2 |
| Hawk-7B | 300 | 80.0 | 74.4 | 45.9 | 77.6 | 69.9 | 35.0 | 63.8 | 69.6 |
| Griffin-7B | 300 | 81.0 | 75.4 | 47.9 | 78.6 | 72.6 | 39.3 | 65.8 | 71.1 |
| **(Hybrid)** | | | | | | | | | |
| StripedHyena-Nous-7B | - | 78.8 | 77.2 | 40.0 | 76.4 | 66.4 | 26.0 | 60.8 | 67.8 |
| Zamba-7B | 1000 | 81.4 | 74.5 | 46.6 | 80.2 | 76.4 | 57.7 | 69.5 | 71.8 |
| Zamba2-7B | 2100 | 81.0 | 80.3 | 56.4 | 81.5 | 77.2 | 64.8 | 73.5 | 75.3 |
| **(Linearized)** | | | | | | | | | |
| Liger-GLA-Llama-3-8B (Ours) | 0.02 | 80.3 | 81.1 | 52.5 | 76.3 | 72.0 | 43.4 | 67.6 | 72.4 |
| Liger-GLA-Llama-3-8B-H (Ours) | 0.02 | 80.6 | 80.7 | 52.7 | 76.9 | 71.4 | 44.4 | 67.8 | 72.5 |

*Table 3.* **Performance Comparison of Pre-trained and Linearized LLMs on Common-sense Reasoning and Knowledge Benchmarks.** Results span Transformer-based (Mistral-7B, Llama-3-8B), linear/subquadratic (Mamba, RWKV), hybrid (Zamba), and our linearized Liger-GLA variants on language modeling and understanding tasks. Our Linearized Liger models achieve competitive performance with only 0.02B training tokens, demonstrating efficient adaptation to gated linear recurrent architectures.

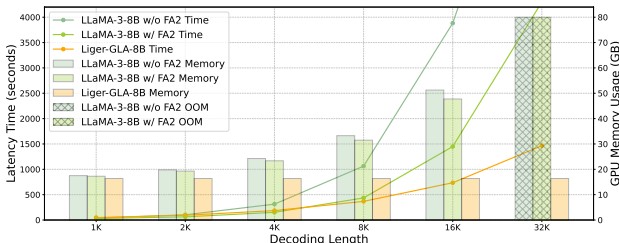

*Figure 4.* **Decoding Latency Time and GPU Memory Usage of Each 8B Models.** We variate the decoding length from 1K to 32K with fixed batch size of 16 on single A800 80GB GPU to evaluate the models' efficiency. Liger enjoys linear-time inference with constant GPU memory usage.

## 4.2. Main Results: Liger can recover pre-trained LLMs' performance more effectively ($\mathcal{RQ}1$)

To validate the effectiveness of our proposed method, we conducted experiments on a series of language modeling and understanding tasks, including PiQA (Bisk et al., 2020), ARC-easy (ARC-e), ARC-challenge (ARC-c) (Clark et al., 2018), HellaSwag (Hella.) (Zellers et al., 2019), Wino-Grande (Wino.) (Sakaguchi et al., 2019) and MMLU (Li et al., 2023). The results are reported in Table 2 and all evaluations were done using lm-evaluation-harness (Gao et al., 2024). Liger, utilizing only 0.02B tokens, achieves a linear recurrent model that recovers 90% of Mistral's performance and 93% of Llama-3's performance with only 0.085% model parameters LoRA finetuning. Our method significantly outperforms other linearization baselines, including SUPRA (Mercat et al., 2024) and Mamba In Llama (Wang et al., 2024), which still need billions of tokens for linear

| Model | Avg. | Avg. |
|---|---|---|
| | ↑ | (no MMLU) ↑ |
| Llama-3.2-1B | 55.1 | 59.9 |
| GLA-1B | 46.9 | 51.1 |
| LoLCATs-Llama-3.2-1B | 51.1 | 56.7 |
| Liger-GLA-Llama-3.2-1B | 52.9 | 59.0 |
| Llama-3.2-3B | 66.1 | 68.1 |
| GLA-3B | 49.1 | 53.8 |
| LoLCATs-Llama-3.2-3B | 55.6 | 62.0 |
| Liger-GLA-Llama-3.2-3B | 60.7 | 66.5 |
| Llama-3-8B | 71.7 | 73.0 |
| LoLCATs-Llama-3-8B | 62.2 | 70.0 |
| Liger-GLA-Llama-3-8B (Ours) | 67.6 | 72.4 |

*Table 4.* **Scalability Analysis of Linearized Llama-3 Architectures across Model Sizes (1B to 8B).** Liger demonstrates consistent scaling laws, outperforming LoLCATs by +6.8–11.5% absolute on average metrics while preserving 83–98% of base model capabilities with only 0.02B adaptation tokens.

recurrent architecture conversion fine-tuning, and LoLCATs' linearization approach, which requires a two-stage process and twice the number of training tokens.

We also compared Liger with other pre-trained models, including the Transformer models Mistral (Jiang et al., 2023) and Llama-3 (Touvron et al., 2023), linear/subquadratic models such as Mamba (Gu & Dao, 2023), RWKV-6 (Peng et al., 2023), TransNormerLLM (Qin et al., 2023), Hawk and Griffin (De et al., 2024), as well as hybrid models like StripedHyena (Poli et al., 2023), Zamba and Zamba2 (Glorioso et al., 2024). As shown in Table 3, our proposed method outperformed nearly all of the pre-trained

| Gated Linear Recurrent Variants | Gated Memory Formulation | Output Formulation | Form of Gate G | Avg. | MMLU |
|---|---|---|---|---|---|
| | | | | 0-shot | 5-shot |
| Liger-GLA | $\mathbf{S}_t = \mathbf{G}_t \odot \mathbf{S}_{t-1} + \boldsymbol{k}_t^\top \boldsymbol{v}_t$ | $\boldsymbol{o}_t = \boldsymbol{q}_t \mathbf{S}_t$ | $\mathbf{G}_t \in \mathbb{R}^D$ | 72.4 | 43.4 |
| Liger-HGRN2 | $\mathbf{S}_t = \mathbf{G}_t \mathbf{S}_{t-1} + (1 - \mathbf{G}_t)^\top \boldsymbol{v}_t$ | $\boldsymbol{o}_t = \boldsymbol{q}_t \mathbf{S}_t$ | $\mathbf{G}_t \in \mathbb{R}^D$ | 69.5 | 36.2 |
| Liger-GSA | $\begin{cases} \tilde{\mathbf{K}}_t = \mathbf{G}_t \tilde{\mathbf{K}}_{t-1} + (1 - \mathbf{G}_t)^\top \boldsymbol{k}_t \\ \tilde{\mathbf{V}}_t = \mathbf{G}_t \tilde{\mathbf{V}}_{t-1} + (1 - \mathbf{G}_t)^\top \boldsymbol{v}_t \end{cases}$ | $\boldsymbol{o}_t = \tilde{\mathbf{V}}_t \operatorname{Softmax}(\tilde{\mathbf{K}}_t^\top \boldsymbol{q}_t)$ | $\mathbf{G}_t \in \mathbb{R}^M$ | 70.5 | 41.2 |

*Table 5.* **Gated Linear Recurrent Model Variants with Liger**. Liger can be applied to the efficient linearization of various linear recurrent structures with gating mechanism and achive high quality performance recovery.

linear/subquadratic models and achieve competitive performance compared with transformer-based and hybrid LLMs.

### 4.3. Liger is a Efficient and Scalable Hybrid Structure ($\mathcal{RQ}$2 & $\mathcal{RQ}$3)

We conducted experiments to compare efficiency in terms of decoding latency speed and GPU memory consumption of Llama-3-8B without (w/o.) Flash-Attention-2 (FA2), Llama-3-8B with (w/.) Flash-Attention-2 (FA2) and Liger-GLA-8B on single A800 80GB GPU. We set a fixed batch size of 16 and variate the decoding sequence length from 1K to 32K with the fixed prefix input length of 128. As presented in Fig. 4, we observe that Liger achieves linear-time decoding complexity while maintaining constant memory usage.

We evaluate efficiency across three scales of the Llama-3 series (1B, 3B, 8B), comparing vanilla transformers, GLA, LoLCATs, and our Liger-GLA. As shown in Table 4, Liger consistently outperforms both GLA and LoLCATs while preserving 93% of the base Llama-3's performance on average. Notably, GLA-1B substantially underperforms all methods (46.9% average), highlighting the necessity of our parameter-efficient adaptation strategy. The performance gap between Liger and vanilla Llama-3 narrows with model size ($\Delta = 4.8\%$ at 1B $\rightarrow \Delta = 1.8\%$ at 8B), indicating improved architectural compatibility at scale.

We conduct experiments on gated linear recurrent structure variations including GLA (Yang et al., 2023), HGRN2 (Qin et al., 2024c) and GSA (Zhang et al., 2024d), with the results presented in Table 5, demonstrating the extensibility of Liger on various gated linear recurrent structures.

### 4.4. Liger Framework Analysis ($\mathcal{RQ}$4)

To verify the key components of Liger, we conducted ablation studies on the model structure. Specifically, we experimented with using gates generated from randomly initialized gate projections (Gate Proj.) instead of pooling, and adopting pure linear attention without considering gating mechanism (Pure LA). Additionally, we incorporated learnable feature map modules (Feat. Map.), similar to Zhang et al. (2024b). We also evaluated the effects of removing LoRA (w/o LoRA), GLA (w/o GLA) and SWA (w/o SWA)

| Model Variants | Validation PPL. | Avg. | Avg. |
|---|---|---|---|
| | $\downarrow$ | $\uparrow$ | (no MMLU) $\uparrow$ |
| Liger-GLA | 2.96 | 67.6 | 72.4 |
| - Gate Proj. | 3.16 | 63.8 | 68.8 |
| - Feat. Map. | 9.04 | 43.5 | 40.2 |
| - Pure LA | 3.00 | 66.1 | 71.5 |
| - w/o LoRA | 3.23 | 61.7 | 68.1 |
| - w/o SWA | 3.75 | 54.2 | 60.2 |
| - w/o GLA | 3.01 | 66.2 | 72.0 |

*Table 6.* **Ablation Study of Liger on Gated Linear Attention.** We linearize Llama-3-8B into Gated Linear Attention (GLA) to evaluate the key components of Liger. We report Validation perplexity (PPL.) on cleaned alpaca dataset after Liger linearization and the average performance on language modeling and understanding tasks.

individually. The results of these experiments are detailed in Table 6, demonstrating the effectiveness of proposed components in Liger.

## 5. Related Work

**Linear Recurrent Models and their Hybrid.** To address the challenges of quadratic complexity computation cost in standard softmax attention, many linear recurrent models are proposed to achieve efficient training and inference. Data-dependent gating/decay has been proved as an effective mechanism to control the memory changes and improve sequence modeling expressiveness (Yang et al., 2023; Peng et al., 2024; Qin et al., 2024c; Zhang et al., 2024c; Du et al., 2025). Recently, some layer-wise hybrid model architectures have been proposed to compensate for the lack of memory capacity in the linear recurrent models, such as StripedHyena-Nous (Poli et al., 2023), Jamba (Lieber et al., 2024), Zamba (Glorioso et al., 2024), Hymba (Dong et al., 2024), Titans (Behrouz et al., 2024) and Minimax-01 (MiniMax et al., 2025). Actually, all of these hybrid attention architectures introduce extra modules (e.g. feature mapping or gate modules) that need to be trained from scratch, which increases the complexity of the model architecture design and may lead to suboptimal softmax attention approxima-

tion. In addition, integration of softmax and linear attention shows great potential as a new intra-layer hybrid attention paradigm, such as Agent Attention (Han et al., 2024), Based (Arora et al., 2024), GSA (Zhang et al., 2024c). However, these inter-layer hybrid attention forms have linear recurrent modules that only focus on long-term modeling and ignore local information (Arora et al., 2024), and lack a gating mechanism and the approximation of the attention output distribution by controlling the hybrid ratio (or hybrid weights) (Han et al., 2024; Arora et al., 2024), which is particularly critical in the linearization process.

**Linearizing Large Language Models.** Linearizing or fine-tuning (uptraining) transformers to linear-RNNs could significantly reduce the cost of training a brand-new large-scale linear recurrent model architecture by distilling knowledge from pre-trained LLMs. Most linearization methods are proposed to uptrain transformer-based LLMs into linear-RNNs by introducing extra feature mapping modules (Kasai et al., 2021; Mercat et al., 2024; Chen et al., 2024) or adding a loss (Zhang et al., 2024b; Bick et al., 2024; Zhang et al., 2024a) to approximate softmax attention. However, these methods have to introduce extra modules that cannot reuse the existing pre-trained LLM weights and need to be trained from scratch, which may not match the output of the original attention and increases the complexity of the model architecture, leading to suboptimal attention approximation.

## 6. Conclusion

This paper introduces **Liger**, a novel method for linearizing Transformer-based LLMs into gated linear recurrent structures. By leveraging the key matrix weights of pretrained standard-structure models and repurposing them to construct the gating mechanisms, Liger avoids the need for additional parameters and extensive fine-tuning, making it a cost-effective and efficient linearization approach. The use of end-to-end LoRA fine-tuning restores model performance while minimizing linearization costs. Furthermore, the introduction of Liger Attention enhances nearly 93% performance recovery with only 0.02% pre-training tokens, making Liger a competitive solution across a range of language modeling and understanding tasks. Our results demonstrate the effectiveness of Liger on models ranging from 1B to 8B parameters, offering a promising path toward more efficient deployment of large-scale LLMs with linear-time inference and constant memory usage.

## Acknowledgements

This work is supported by the Shanghai AI Laboratory.

## Impact Statement

This work represents a notable advancement in artificial intelligence and machine learning, particularly in linearizing the pretrained Transformer-based models into gated recurrent structures. Liger enables the processing of much longer sequences compared to existing methods while significantly accelerating computation, making it highly beneficial for tasks like natural language understanding, genomic sequence analysis, and time-series forecasting. However, the enhanced capabilities and efficiency introduced by Liger also raise ethical and societal considerations, such as the potential for misuse in generating persuasive but misleading content or in surveillance applications. Nevertheless, the contributions of Liger to reducing computational overhead and energy consumption in training large models may also bring positive environmental impacts.

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

## A. Datasets and Benchmarks

We linearize Liger on Cleaned Alpaca dataset (Taori et al., 2023) and evaluate on downstream language tasks using lm-evaluation-harness (Gao et al., 2024).

- **Cleaned Alpaca** (Taori et al., 2023): The cleaned Alpaca dataset is a structured dataset designed for instruction-tuning of language models, containing 52,000 instructions and corresponding outputs generated by OpenAI's text-davinci-003 engine. Each entry in the dataset includes an "instruction" field that specifies the task for the model, an optional "input" field providing context or additional information, and an "output" field with the model's response. The dataset is formatted in JSON and is intended to enhance the ability of language models to follow instructions effectively.

- **PiQA** (Bisk et al., 2020): The PiQA (Physical Interaction: Question Answering) dataset is designed to assess physical commonsense reasoning, containing 3,084 samples for testing. Each instance includes a "goal" field representing the question, two "solution" fields with potential answers, and a "label" indicating the correct solution. The dataset focuses on everyday situations requiring physical commonsense.

- **ARC-Easy & ARC-Challenge** (Clark et al., 2018): The ARC (AI2 Reasoning Challenge) dataset is a collection of 7,787 genuine grade-school level multiple-choice science questions. It is divided into two subsets: ARC-Easy and ARC-Challenge. The ARC-Easy subset contains relatively straightforward questions that test basic knowledge, while the ARC-Challenge subset includes more complex and difficult questions that require advanced reasoning abilities

- **HellaSwag** (Zellers et al., 2019): The HellaSwag dataset is a comprehensive collection of narrative reasoning tasks designed to evaluate a model's ability to predict the next event in a sequence. It consists of 10,125 examples, each containing a context and four possible endings, with one correct and three incorrect options. The dataset is derived from the ActivityNet Captions corpus and is structured to test the model's understanding of narrative coherence and common-sense reasoning based on the given context.

- **WinoGrande** (Sakaguchi et al., 2019): The WinoGrande dataset is a large-scale collection of 44,000 problems designed to evaluate commonsense reasoning, inspired by the Winograd Schema Challenge but enhanced in scale and difficulty. Each problem is structured as a fill-in-the-blank task with two options, requiring models to choose the correct answer based on contextual understanding. WinoGrande is significantly more challenging than the original Winograd Schema Challenge, with state-of-the-art models achieving lower accuracy, highlighting its effectiveness in testing true commonsense understanding.

- **MMLU** (Li et al., 2023): The MMLU (Massive Multitask Language Understanding) dataset is a comprehensive benchmark designed to evaluate AI models' general knowledge across a wide range of subjects and languages. It comprises 57 distinct categories, spanning elementary-level knowledge to advanced professional topics such as law, physics, history, and computer science. The dataset has been translated into 14 languages using professional human translators, ensuring high-quality and accurate translations. This multilingual approach aims to improve the inclusivity and effectiveness of AI models across different linguistic communities.

## B. Experiment Details

Our 8B model linearization experiments are conducted on single NVIDIA A800 80G GPU. With batch size 1 and gradient accumulation over 8 batches, Liger method takes around 4 hours and 27GB GPU memory usage for 2 epochs end-to-end linearization, instead of any multi-stage training. All our experiments were conducted and evaluated using a fixed random seed of 0 to ensure reproducibility.

## C. Model Weight Changes

We analyze the model weight changes after model architecture linearization. Let $W$ be the sum of each weight parameter of LLM, we calculate the model weight changes $\Delta W$ in Liger compared with LoLCATs. As presented in Table 7, we observe that Liger has a smaller number of changing parameters than LoLCATs, which mitigates catastrophic forgetting of pretrained knowledge during linearization process, thereby enhancing both training efficiency and model effectiveness.

|  | LoLCATs | Liger |
|---|---|---|
| $\parallel \Delta W \parallel_1$ | 3.68e+06 | 2.91e+06 |
| $\parallel \Delta W \parallel_1 / W$ | 4.82% | 3.85% |
| $\parallel \Delta W \parallel_2$ | 3.93e+04 | 1.27e+03 |
| $\parallel \Delta W \parallel_2 / W$ | 179.29% | 7.63% |

*Table 7.* **Model Weight Changes after Linearization.** Liger has a smaller number of changing model weights than LoLCATs.

| Model | PiQA | ARC-e | ARC-c | Hella. | Wino. | MMLU | Avg. | Avg. |
|---|---|---|---|---|---|---|---|---|
| | acc ↑ | acc ↑ | acc_norm ↑ | acc_norm ↑ | acc ↑ | acc (5-shot) ↑ | ↑ | (no MMLU) ↑ |
| Liger-GLA-8B-Q-Pooling | 80.2 | 78.6 | 51.6 | 76.7 | 71.6 | 41.7 | 66.7 | 71.7 |
| Liger-GLA-8B-K-Pooling | 80.3 | 81.1 | 52.5 | 76.3 | 72.0 | 43.4 | 67.6 | 72.4 |
| Liger-GLA-8B-V-Pooling | 80.7 | 77.4 | 50.6 | 76.4 | 70.4 | 43.7 | 66.5 | 71.1 |

*Table 8.* **Results on Gate Construction from Different Components.** We compare the Liger-GLA-8B's performance of gate module obtained by pooling from query (Q), key (K) and value (V) matrices, the gate construction from key matrix outperforms others.

## D. Gate Construction from Different Components

We consider constructing gating mechanisms from different components of the model. We compare the Liger-GLA-8B's performance of gate module obtained by pooling from query (Q), key (K) and value (V) matrices, and the results are shown in Table 8. We observed that the gate construction from key matrix outperforms than others. The key matrix usually serves as memory indexing in transformer or linear model (Gershman et al., 2025), which is similar to the gate that determines which part of the memory should be retrieved or forgotten. In this view, it is intuitive to choose Key for gate construction. MetaLA (Chou et al., 2024a) also points out that the linear recurrent model needs to meet the "least parameter approximation" condition to achieve the optimal linear attention design. In this case, the parameters of key matrix are redundant and can be used to construct the gate, which also provides motivation and theoretical support for gate derived from Key.

## E. Full Results

| Sequence Length | Llama-3-8B w/o FA2 | | Llama-3-8B w/ FA2 | | Liger-8B | |
|---|---|---|---|---|---|---|
| | Time | Memory | Time | Memory | Time | Memory |
| 1K | 37.92 | 17.50 | 29.36 | 17.26 | 47.83 | 16.37 |
| 2K | 102.54 | 19.75 | 62.52 | 19.29 | 94.41 | 16.37 |
| 4K | 312.98 | 24.25 | 151.51 | 23.35 | 185.79 | 16.37 |
| 8K | 1062.65 | 33.26 | 436.04 | 31.48 | 367.78 | 16.37 |
| 16K | 3882.36 | 51.26 | 1449.20 | 47.73 | 734.91 | 16.37 |
| 32K | - | OOM | - | OOM | 1465.52 | 16.37 |

*Table 9.* **Detailed Results on Inference Efficiency in terms of Decoding Latency Time and GPU Memory Usage.** We present the decoding latency time (seconds) and the GPU memory usage (GB) during inference stage compared with Llama-3-8B without (w/o), with (w/) Flash-Attention-2 (FA2) and Liger-8B. OOM denotes out of memory.

| Model | PiQA | ARC-e | ARC-c | Hella. | Wino. | MMLU | Avg. | Avg. |
|---|---|---|---|---|---|---|---|---|
| | acc ↑ | acc ↑ | acc_norm ↑ | acc_norm ↑ | acc ↑ | acc (5-shot) ↑ | ↑ | (no MMLU) ↑ |
| Liger-GLA | 80.3 | 81.1 | 52.5 | 76.3 | 72.0 | 43.4 | 67.6 | 72.4 |
| Liger-HGRN2 | 79.2 | 76.8 | 48.5 | 74.4 | 68.8 | 36.2 | 64.0 | 69.5 |
| Liger-GSA | 79.5 | 78.5 | 49.4 | 74.5 | 70.5 | 41.2 | 65.6 | 70.5 |

*Table 10.* **Full Results on Different Gated Linear Recurrent Model Variants with Liger.** Liger can be applied to the efficient linearization of various linear recurrent structures with gating mechanism and achieve high quality performance recovery.

| Model | PiQA | ARC-e | ARC-c | Hella. | Wino. | MMLU | Avg. | Avg. |
|---|---|---|---|---|---|---|---|---|
| | acc ↑ | acc ↑ | acc_norm ↑ | acc_norm ↑ | acc ↑ | acc (5-shot) ↑ | ↑ | (no MMLU) ↑ |
| Llama-3.2-1B | 74.1 | 65.4 | 36.4 | 63.8 | 60.0 | 31.0 | 55.1 | 59.9 |
| GLA-1B | 69.9 | 55.2 | 27.6 | 48.9 | 53.9 | 25.9 | 46.9 | 51.1 |
| LoLCATs-Llama-3.2-1B | 74.1 | 63.7 | 36.4 | 51.2 | 58.2 | 23.1 | 51.1 | 56.7 |
| Liger-GLA-Llama-3.2-1B | 75.0 | 65.4 | 35.7 | 59.8 | 59.1 | 22.4 | 52.9 | 59.0 |
| Llama-3.2-3B | 76.4 | 74.7 | 46.0 | 73.6 | 69.9 | 56.2 | 66.1 | 68.1 |
| GLA-3B | 71.5 | 59.2 | 30.0 | 53.0 | 55.3 | 25.6 | 49.1 | 53.8 |
| LoLCATs-Llama-3.2-3B | 76.7 | 72.0 | 42.3 | 51.9 | 66.9 | 23.6 | 55.6 | 62.0 |
| Liger-GLA-Llama-3.2-3B | 77.9 | 74.0 | 43.9 | 70.3 | 66.3 | 32.1 | 60.7 | 66.5 |
| Llama-3-8B | 79.4 | 80.1 | 53.2 | 79.2 | 72.9 | 65.3 | 71.7 | 73.0 |
| LoLCATs-Llama-3-8B | 80.1 | 80.4 | 53.5 | 63.4 | 72.9 | 42.8 | 65.5 | 70.0 |
| Liger-GLA-Llama-3-8B (Ours) | 80.3 | 81.1 | 52.5 | 76.3 | 72.0 | 43.4 | 67.6 | 72.4 |

*Table 11.* **Full Results on Scalability Analysis of Linearized Llama-3 Architectures across Model Sizes (1B to 8B).** Performance comparisons between vanilla Llama-3, GLA, LoLCATs, and our Liger-GLA variants on language modeling and reasoning tasks. Liger demonstrates consistent scaling laws, outperforming LoLCATs by +6.8–11.5% absolute on average metrics while preserving 83–98% of base model capabilities with only 0.02B adaptation tokens.

| Model | Validation PPL. | PiQA | ARC-e | ARC-c | Hella. | Wino. | MMLU | Avg. | Avg. |
|---|---|---|---|---|---|---|---|---|---|
| | ↓ | acc ↑ | acc ↑ | acc_norm ↑ | acc_norm ↑ | acc ↑ | acc (5-shot) ↑ | ↑ | (no MMLU) ↑ |
| Liger-GLA | 2.96 | 80.3 | 81.1 | 52.5 | 76.3 | 72.0 | 43.4 | 67.6 | 72.4 |
| - Gate Proj. | 3.16 | 79.1 | 75.9 | 49.6 | 71.8 | 67.3 | 39.2 | 63.8 | 68.8 |
| - Feat. Map. | 9.04 | 63.1 | 46.3 | 24.2 | 33.7 | 50.4 | 23.8 | 40.2 | 43.5 |
| - Pure LA | 3.00 | 79.9 | 79.8 | 52.0 | 75.3 | 70.6 | 38.8 | 66.1 | 71.5 |
| - w/o LoRA | 3.23 | 78.7 | 75.6 | 47.4 | 74.0 | 64.8 | 29.5 | 61.7 | 68.1 |
| - w/o SWA | 3.75 | 75.0 | 68.3 | 39.1 | 63.4 | 55.3 | 26.4 | 54.2 | 60.2 |
| - w/o GLA | 3.01 | 79.8 | 80.5 | 52.4 | 76.3 | 72.4 | 37.0 | 66.2 | 72.0 |

*Table 12.* **Full Results on Ablation Study.** We linearize Llama-3-8B to Gated Linear Attention (GLA) to evaluate the key components of Liger. We report Validation perplexity (PPL.) on cleaned alpaca dataset after Liger linearization and the average performance on language modeling and under-standing tasks.

