# OpenReview forum: "Liger: Linearizing Large Language Models to Gated Recurrent Structures"
_ICML.cc/2025/Conference — ICML 2025 poster_

### Official Review · Reviewer_5t3M · 2025-03-14

**Overall Recommendation:** 4

**Summary:**

Transformer Language models based on linear recurrent structures  (Katharopoulos et al., 2020; Yang et al., 2023; Qin et al., 2024b) are substantially more efficient than regular transformers due to their linear dependency on sequence length and constant memory requirements. Prior work - SUPRA (Mercat et al., 2024), MambaInLlama (Wang et al., 2024) and LoL-CATs (Zhang et al., 2024a) - has shown that is is possible to initialize these models from existing transformer based LLMs and linearize them. However, this process requires costly fine-tuning steps and results in loss of performance. One solution to improve performance is via introducing a gating operation in the recurrence. This paper presents an approach, called Liger, to speedup the model transfer and improve performance by repurposing existing key vectors for the recurrent gating operation. Lora fine-tuning is employed to fine-tune these parameters. The paper also presents an approach to interpolate gated-recurrent attention with a sliding-window softmax attention.  The liger attention block is used along with the original transformer attention block in the model. Results are presented on Mistral-7B and Llama3-8B llms. On Mistral-7B, the approach outperforms SUPRA trained on 500x training tokens and Lolcat which is trained on a similar number of training tokens. On Llama3-8b, the approach outperforms Lolcat trained on a similar number of training tokens and Supra trained on 100x training tokens and some versions of Mamba trained on 100x training tokens. However it underperforms a variant of Mamba trained on 100x training tokens with 50% attention. It achieves a linear time decoding speed when examined on sequences ranging from  1k to 32k. Ablations are shown on each of the critical components such as gates generated from random projections. Lora, feature map, SWA.

## update after rebuttal
The authors have addressed the clarity issues that I raised in my initial review. Hence increasing the scores after the rebuttal.

**Claims And Evidence:**

The claims are supported by evidence.

**Essential References Not Discussed:**

None

**Experimental Designs Or Analyses:**

Yes, the designs are sound.

**Methods And Evaluation Criteria:**

Yes they make sense.

**Other Comments Or Suggestions:**

* Figure 4: It would be useful to show results from GLA and LolCat as well.
* L41: resurrent -> recurrent
* L84: rely -> reliant

**Other Strengths And Weaknesses:**

Strengths:
* Presents an approach to speed up the conversion of a transformer based LLM into linear recurrent structures and improve performance using gating.
* Reports results on Mistral-7B and Lllama3-8B showing that the approach yields improvements in performance compared to prior linearization approaches while training on similar or substantially fewer tokens.
* Presents an ablation study showing the importance of each of the components added.

Weaknesses:
* Some parts of the paper are unclear. See questions below.

**Questions For Authors:**

* L208: Eqn 6: Is pooling performed across the time dimension? Please clarify.
* L273: 'GLA denotes Gated Recurrent Modeling' -> should GLA be GRM instead?
* Table 2/4: What does Liger-GLA mean? Does this not include Sliding window attention?
* L417: What do learnable feature map modules mean? It would be good to explain briefly.

**Relation To Broader Scientific Literature:**

Language models based on linear recurrent structures  (Katharopoulos et al., 2020; Yang et al., 2023; Qin et al., 2024b) are substantially more efficient than transformers due to their linear dependency on sequence length and constant memory requirements. Prior work - SUPRA (Mercat et al., 2024), MambaInLlama (Wang et al., 2024) and LoL-CATs (Zhang et al., 2024a) - has shown that is is possible to reuse existing transformer based LLMs by linearizing them. However, this process requires costly fine-tuning steps and results in loss of performance. One solution to improve performance is via introducing a gating operation in the recurrence. This paper presents an approach, called Liger, to speedup the model transfer and improve performance by repurposing existing key vectors for the recurrent gating operation. Lora fine-tuning is employed to fine-tune these parameters. The paper also presents an approach to interpolate gated-recurrent attention with a sliding-window softmax attention.

**Theoretical Claims:**

No theoretical claims are made in the paper.

---

> ### Author Rebuttal · Authors · 2025-03-31
>
> Dear Reviewer 5t3M,
>
> Thanks for your thorough reviews and we appreciate the attention to detail and would fix the writing errors in the revision.  We give point-by-point responses to your comments based on your questions.
>
> > `Q1`: Is pooling performed across the time dimension?
>
> No, the pooling operation is performed accross hidden dimension. We perform linearization by pooling the model's $K \in \mathbb{R}^{T \times D} $ into $G \in \mathbb{R}^{T \times D_G}$, where $D_G$ depends on the gating module of the specific model used, such as $G \in \mathbb{R}^{T \times S}$ in GSA, where $S$ is the number of memory slots. This form can achieve the effect of dynamic gating without introducing any learnable parameters during the linearization process. It is simple and has been demonstrated to be particularly effective in experiments.
>
> > `Q2`: 'GLA denotes Gated Recurrent Modeling' -> should GLA be GRM instead?
>
> Yes, thanks for your careful review. Here GLA should be GRM instead. GRM represents a general form of various gated recurrent modeling methods and their variants, and GLA is an instance of GRM.
>
> > `Q3`: What does Liger-GLA mean? Does this not include Sliding window attention?
>
> Liger-GLA means using Liger proposed in our paper for linearization of LLM to gated recurrent structures, which adopts Liger Attention, a intra-layer hybrid attention including Sliding Window Attention and using GLA as the instance of GRM.
>
> > `Q4`: What do learnable feature map modules mean?
>
> The original softmax attention is calculated as $O=\text{softmax}(QK)V$, and its complexity is $\mathbb{O}(T^2)$ quadratic over the length of the sequence. Linear attention is achieved by removing the softmax operation and replacing it with **feature map modules** $\phi$ acting on $Q$ and $K$ and reducing the computational complexity to $\mathbb{O}(T)$ through the right multiplication technique. The specific formula is $O=\phi(Q)\phi(K)V$, so that the product of $\phi(Q)\phi(K)$ can approximate the output of softmax attention. For example, [1] setting feature map to $\phi(x)=\text{elu}(x)+1$ introduces non-negativity similar to softmax attentioin to achieve better performance.
>
> **Learnable feature map modules** introduce learnable parameter for $\phi$ optimization, which makes the model adaptive to data and improves expressiveness to better approximate the effect of softmax attention. For example, the feature map module design in [2] adopts: $\phi(x)=\exp(\mathbf{W}^\top x+b)$.
>
> In our proposed Liger, we set $\phi(x)$ which simply uses the normalization function, which can ensure that the distribution of the product of $\phi(Q)\phi(K)$ is still non-negative and approximates the original $\text{softmax}(QK)$ output distribution, and there is no need to introduce any learnable parameters, which is very helpful for LLM linearization and demonstrated effective in our experiments.
>
> ___
>
> [1] Angelos Katharopoulos, et al. Transformers are RNNs: Fast Autoregressive Transformers with Linear Attention. ICML, 2020. \
> [2] Michael Zhang, et al. The Hedgehog & the Porcupine: Expressive Linear Attentions with Softmax Mimicry. ICLR, 2024.

---

### Official Review · Reviewer_2WYJ · 2025-03-16

**Overall Recommendation:** 3

**Summary:**

This paper proposed a novel method, Liger, for transforming pretrained Transformer-based LLMs into gated linear recurrent models, repurposing key matrix weights to create gating mechanisms.
It utilizes Liger Attention, an intra-layer hybrid attention mechanism involving Gated Recurrent Modeling (GRM) and Sliding Window Attention (SWA). The resulting model retains 93% of the original Transformer model’s performance while requiring only 0.02% of the pretraining tokens.
Using lightweight LoRA finetuning, Liger successfully restores the performance of linearized models, achieving state-of-the-art efficiency and accuracy across multiple benchmarks on models ranging from 1B to 8B parameters.

**Claims And Evidence:**

- This work asserts that the proposed method can adapt a pretrained transformer-based LLM into a gated recurrent structure with high token efficiency. This assertion is well supported by the method and empirical results.
- This work asserts that the proposed Liger Attention is an intra-layer hybrid attention mechanism that combines sliding window softmax attention with linear recurrent modeling. This assertion is well supported by the method and empirical results.

**Essential References Not Discussed:**

One of the contributions is the intra-layer hybrid attention, referred to as Liger Attention. A similar design is also found in [1,2].

[1] Dong, X., Fu, Y., Diao, S., Byeon, W., Chen, Z., Mahabaleshwarkar, A.S., Liu, S.Y., Van Keirsbilck, M., Chen, M.H., Suhara, Y. and Lin, Y., 2024. Hymba: A hybrid-head architecture for small language models. arXiv preprint arXiv:2411.13676.
[2] Behrouz, A., Zhong, P. and Mirrokni, V., 2024. Titans: Learning to memorize at test time. arXiv preprint arXiv:2501.00663.

**Experimental Designs Or Analyses:**

Yes

**Methods And Evaluation Criteria:**

This work evaluates the proposed method regarding task accuracy and latency.

**Other Comments Or Suggestions:**

N/A

**Other Strengths And Weaknesses:**

see Questions

**Questions For Authors:**

- It appears that all methods experience significant degradation in performance on the MMLU task, including the "Mamba2-Llama-3-8B 50% Attn." model. It would be beneficial to conduct further analysis to understand why "Liger-GLA-Llama-3-8B" does not perform well on MMLU as well.

- Why does the gate need to be derived from the Key instead of other elements?

- How much do the LoRA weights change after fine-tuning?

**Relation To Broader Scientific Literature:**

N/A

**Theoretical Claims:**

N/A

---

> ### Author Rebuttal · Authors · 2025-03-31
>
> Dear Reviewer 2WYJ,
>
> We are truly grateful for the time you have taken to review our paper and your insightful comment. We address your questions in the following.
>
> > `Q1`: Discussion on intra-layer hybrid attention with similar designs
>
> We carefully read these works for further comparison and discussion.
>
> Hymba uses a intra-layer hybrid attention which combines softmax attention and SSM via independent parameterizations: $o_t=\beta_1\text{norm}(M_{attn}x_t)+\beta_2\text{norm}(M_{ssm}x_t)$, where $M_{attn}$ and $M_{ssm}$ employ separate projection matrices $W_Q/W_K/W_V$ vs. $W_A/W_B/W_C$. Liger Attention eliminates this duplication by: $o_t=\alpha\text{GRM}(q_t,k_t,v_t)+\beta\text{SWA}(q_t,k_t,v_t)$, which shares matrices between sliding window attention (SWA) and gated recurrent modules (GRM), avoiding the need for introducing extra learnable parameters. This parameter efficiency directly supports better linearization while retaining generality—GRM subsumes (but is not limited to) SSM.
>
> Titans' hybrid form also similarly uses independent parameters to construct linear and attention modules respectively, and is essentially an inter-layer hybrid rather than intra-layer, because the output of its memory module (linear module) will be used as the input of the attentnion module, and vice versa.
>
> We would respectfully cite and supplement the discussion of related works.
>
> > `Q2`: Significant degradation in performance on MMLU
>
> We have noticed that most linearization methods have significant degradation on MMLU. We hypothesize that the performance gap partially stems from MMLU's evaluation design, which assesses both knowledge recall in pretrained weights and context-based answer retrieval through its 5-shot multiple-choice format[4]. While this setup requires models to identify correct answer indices from contextual prompts, prior research demonstrates that linear recurrent architectures exhibit **inherent limitations** in such retrieval tasks[5].
>
> Despite this, we still try our best to alleviate this problem. We  linearize Liger from Llama-3-8B-instruct with stronger instruction following ability. Besides, we investigate whether we can improve MMLU performance from stronger LLM (e.g. Qwen2.5).
>
> ||MMLU|
> |-|-|
> |Llama-3-8B|65.3|
> |Liger-GLA-8B|43.4|
> |Llama-3-8B-Instruct|65.7|
> |Liger-GLA-8B-Instruct|47.6|
> |Qwen2.5-7B|74.2|
> |Liger-GLA-Qwen2.5-7B|55.1|
>
> Although there is still a significant drop on MMLU, we find that Liger can benefit from the instruct model to improve 5-shot performance, and can also benefit from more powerful model (Qwen2.5) for such task (only 1.6 lower than Mamba2-Llama3-8B 50% Attn), further narrowing the gap with the original LLM.
>
> > `Q3`: Why does the gate need to be derived from the Key instead of other elements
>
> The key matrix usually serves as memory indexing in transformer or linear model[1], which is similar to the gate that determines which part of the memory should be retrieved or forgotten. In this view, it is intuitive to choose Key for gate construction. MetaLA[2] also points out that the linear recurrent model needs to meet the "least parameter approximation" condition to achieve the optimal linear attention design. In this case, the key matrix is ​​redundant and can be used to construct the gate, which also provides motivation and theoretical support for gate derived from Key.
>
> We have supplemented experimental comparisons of gate derived from Q and V:
>
> ||Avg|Avg(no MMLU)|
> |-|-|-|
> |Liger-Q-pool|66.7|71.7|
> |Liger-K-pool|67.6|72.4|
> |Liger-V-pool|66.5|71.1|
> |Liger-Gate-init|63.8|68.8|
>
> We can observe that the effect of constructing the gate module from any Q, K, V is ​​significantly better than randomly initialized gate, and the overall performance of gate derived from Key is ​​more outstanding, which is consistent with the conclusion in [2][3] that the parameters of the Key Matrix are more redundant and can act as gating mechanism.
>
> > `Q4`: How much do the LoRA weights change
>
> LoRA is only used to finetune QKV that only 0.059% of total weights need to be trained. Let $W$ be the sum of each weight parameter of LLM, we calculate the change in Liger's LoRA weights compared with LoLCATs:
>
> ||LoLCATs|Liger|
> |-|-|-|
> |$\|\|\Delta W\|\|_1$|3.68e+06|2.91e+06|
> |$\frac{\|\|\Delta W\|\|_1}{W}$|4.82%| 3.85%|
> |$\|\|\Delta W\|\|_2$|3.93e+04|1.27e+03 |
> |$\frac{\|\|\Delta W\|\|_2}{W}$|179.29%|7.63%|
>
> We observe that Liger has a smaller number of changing parameters than LoLCATs, which makes Liger less pretrained knowledge destruction, more efficient and effective.
>
> ___
>
> [1] Gershman S. J., et al. Key-value memory in the brain. arXiv, 2025 \
> [2] Chou Y., et al. MetaLA: Unified Optimal Linear Approximation to Softmax Attention Map. NeurIPS, 2024 \
> [3] Qin Z., et al. HGRN2: Gated Linear RNNs with State Expansion. COLM, 2024 \
> [4] Hendrycks D., et al. Measuring Massive Multitask Language Understanding. arXiv, 2020 \
> [5] Waleffe R., et al. An Empirical Study of Mamba-based Language Models. arXiv, 2024

---

> > ### Comment · Reviewer_2WYJ · 2025-04-01
> >
> > I appreciate the authors' response, which addressed most of my questions.
> > For the MMLU accuracy degradation, I understand that some linear models may mess up answer indices (e.g., A B C D in four-choice questions). I was wondering whether Liger has the same issue. This is very important to understand the benefit of Liger (converted from transformer models) against pre-training a linear attention model from scratch.

---

> > > ### Author Response · Authors · 2025-04-02
> > >
> > > Thank you for your acknowledgment of our rebuttal! We try to answer your question regarding "*whether Liger has the same issue as the linear model*".
> > >
> > > In the setting of the MMLU 5-shot task, the input format is as follows (we show an example):
> > >
> > > ```
> > > The following are multiple choice questions (with answers) about anatomy.
> > >
> > > What is the embryological origin of the hyoid bone?
> > > A. The first pharyngeal arch
> > > B. The first and second pharyngeal arches
> > > C. The second pharyngeal arch
> > > D. The second and third pharyngeal arches
> > > Answer: D
> > >
> > > Which of these branches of the trigeminal nerve contain somatic motor processes?
> > > A. The supraorbital nerve
> > > B. The infraorbital nerve
> > > C. The mental nerve
> > > D. None of the above
> > > Answer: D
> > >
> > > The pleura
> > > A. have no sensory innervation.
> > > B. are separated by a 2 mm space.
> > > C. extend into the neck.
> > > D. are composed of respiratory epithelium.
> > > Answer: C
> > >
> > > In Angle's Class II Div 2 occlusion there is
> > > A. excess overbite of the upper lateral incisors.
> > > B. negative overjet of the upper central incisors.
> > > C. excess overjet of the upper lateral incisors.
> > > D. excess overjet of the upper central incisors.
> > > Answer: C
> > >
> > > Which of the following is the body cavity that contains the pituitary gland?
> > > A. Abdominal
> > > B. Cranial
> > > C. Pleural
> > > D. Spinal
> > > Answer: B
> > >
> > > Which of the following statements is true of the pupillary light reflex?
> > > A. Its efferent limb is carried in the optic nerve
> > > B. It is mediated by the inferior colliculi in the midbrain
> > > C. It is a consensual reflex
> > > D. Its afferent limb is carried in the oculomotor nerve
> > > Answer:
> > > ```
> > >
> > > The model will only output an ABCD option as the answer without any additional information in this 5-shot setting, so we actually cannot know whether the model messes up the answer indices. We tend to believe that there would be some special cases showing that Liger has the similar issues of linear models since Liger still belongs to the linear recurrent model architecture, but Liger can better overcome these defects. The experiment results also demonstrate that Liger is superior to pretraining linear attention models from scratch and other linearization methods.
> > >
> > > Despite this, Liger may still have some inherent limitations of linear models (but its better performance shows that Liger is less affected by these issues). The motivation of Liger is not to completely solve the potential defects of current linear recurrent model architecture. Our contribution is to obtain a more powerful linear model with minimal training cost in resource-constrained scenarios, and its performance is significantly better than the previous pre-trained linear attention models or linearization methods.
> > >
> > > Once again, we sincerely thank you for your thorough review and hope this addresses your concerns! We respectfully look forward to further discussion with you.

---

### Official Review · Reviewer_xf9Q · 2025-03-17

**Overall Recommendation:** 4

**Summary:**

This paper studies the problem of how to convert pretrained standard attention-based LLMs into more efficient hybrid models, a research area that recently emerges along with significant progress on sub-quadratic model architectures. It proposes a methodology to convert current Mistral/Llama models to constant-memory hybrid models that combine sliding window attention and gated recurrent neural networks.

**Claims And Evidence:**

The major claim is that the proposed recipe for obtaining a constant-memory model out of existing pretrained models can recover 93% performance. This claim is clearly supported by the main results.
The efficiency claim of the resulting model is also well supported by benchmarking decoding efficiency in terms of GPU memory usage and latency.

**Essential References Not Discussed:**

NO.

**Experimental Designs Or Analyses:**

While the main results clearly support the major claim of performance gain and efficiency benefit, I believe the experiments should be further improved to understand the core piece of the method. Specifically, it would be great if there were more experiments on the choice of target architecture (i.e., the constant memory model) and the choice of parameterization to address the following concerns:

1. The baseline that only uses sliding window attention is missing, which makes it hard to infer the effect of the linear attention part.
2. The premise of using gated linear attention is that gating is crucial for RNN. However, I believe it's unclear whether gating is still necessary in this distillation setting (i.e., converted from pretrained weights) and hybrid architecture (sliding window + linear attention).
3. Hopefully more discussion on the parameterization of G (e.g., a parametric way as opposed to the current non-parametric way).

**Methods And Evaluation Criteria:**

The central problem towards converting a pretrained LLM to another architecture lies in the parameterization choice, and the proposed method aims to address this problem. The evaluation is based on some standard benchmarks for evaluating LLMs.

**Other Comments Or Suggestions:**

Figure captions should be improved. E.g., table 2 is hard to read with many model variants.

**Other Strengths And Weaknesses:**

No

**Questions For Authors:**

It's unclear to me whether G_t is a scalar or vector. Can you provide details of the pooling operation along the resulting shape of G_t.

**Relation To Broader Scientific Literature:**

This paper connects to the broad research effort on sub-quadratic/efficient model architectures. The results here could also shed light on the connection between softmax-attention and linear attention in terms of parameterization.

**Theoretical Claims:**

NO.

---

> ### Author Rebuttal · Authors · 2025-03-31
>
> Dear Reviewer xf9Q,
>
> We sincerely thank you for your careful comments and thorough understanding of our paper! Here we give point-by-point responses to your comments and questions.
>
> > `Q1`: The baseline that only uses sliding window attention is missing.
>
> Thanks for your careful comment. We have supplemented the experiment with sliding window attention (SWA) only.
>
> ||PiQA|ARC-e|ARC-c|Hella.|Wino.|MMLU|Avg.|Avg.(no MMLU)|
> |-|-|-|-|-|-|-|-|-|
> |GLA only|75.0|68,3|39.1|63.4|55.3|26.4|54.6|60.2|
> |SWA only|79.8|80.5|52.4|75.1|**72.4**|37.0|66.2|72.0|
> |Liger Attention|**80.3**|**81.1**|**52.5**|**76.3**|72.0|**43.4**|**67.6**|**72.4**|
>
> Notably, whether SWA only or GLA only is used, the performance will decrease, while Liger Attention achieves the best performance.
>
> > `Q2`: It's unclear whether gating is still necessary in this distillation setting
>
> While transformer-based LLMs suffer from computational inefficiency due to unbounded historical storage in the KV-cache, linear recurrent models face challenges in dynamically "forgetting" obsolete information without gating mechanism. Prior work in linear recurrent models [1-4] underscores the critical role of gating in enabling adaptive forgetting, suggesting its necessity even in linearization or distillation setting.
>
> However, existing linearization methods often omit gating or introduce randomly initialized gating modules that need extra training, which incur added training costs and suboptimal performance (e.g., Table 6 shows performance drops when learnable gates are added). In contrast, Liger incorporates gating and optimizes its construction without any parameters introduction and achieves better performance, which proves that gating is necessary for linearization.
>
> It is worth noting that although most linearization methods are based on distillation setting and adopt multi-stage training with further fine-tuning. In fact, our proposed Liger **does not** adopt any distillation setting. It uses only **single stage of LoRA fine-tuning** to complete the adaptation of the architecture conversion with only **0.059%** of the parameters need to be fine-tuned on **0.02B** training tokens and restore **over 93%** of the LLM performance. The linearization cost significant reduce and it can work on various linear recurrent architectures with gating mechanism, which reflects the simplicity and versatility of Liger.
>
> > `Q3`: Hopefully more discussion on the parameterization of G
>
> The parametric way for $G$ construction is introducing extra learnable parameters, which dynamically adjusts the flow of information through learnable weights and biases (e.g. MLP projection). It is data-adaptive but requires more computing resources and training data compared with the non-parametric way. In the scenario of LLM linearization, we seek to obtain a powerful linear LLM faster at a lower cost under resource constraints.
>
> Our ablation study (Table 6 in our manuscript) show the following results, where Gate Proj refers to construct the gate projection module in parameteric way by introducing extra parameters, which is implemented as the low-rank MLP used in GLA[1].
>
> ||Valid PPL$\downarrow$|Avg$\uparrow$|Avg(no MMLU)$\uparrow$|
> |-|-|-|-|
> |Liger-GLA|2.96|67.0| 71.7|
> |Parametric Gate Proj.|3.16|63.8|68.8|
>
> The results reveal that Liger can achieve better performance by using a simple pooling operation in non-parameteric way to construct the gating module, which is consistent with the concept discussed above about parameterization of $G$.
>
> > `Q4`: Provide details of the pooling operation along the resulting shape of G_t
>
> The form of $G_t$ depends on the specific gated recurrent model employed. Most gated linear models adopt a dynamic time-varying vector gate instead of a simple scalar. In the specific implementation, our pooling operation uses `torch.AdaptiveAvgPool1d` over the $D$ dimension, which can construct a specified gate form through average pooling or interpolation without introducing any learnable parameters, benefiting the linearization process. For example, the gate of GLA: $G_t \in \mathbb{R}^{D}$ is a vector, while the gate of GSA is $G_t \in \mathbb{R}^{M}$, where $M$ is the number of slots. We can adopt the simple pooling operation to transform the shape from the LLM weight matrix to construct the corresponding gate vector.
>
> ___
>
> [1] Songlin Yang, et al. Gated Linear Attention Transformers with Hardware-Efficient Training. ICML, 2024. \
> [2] Zhen Qin, et al. HGRN2: Gated Linear RNNs with State Expansion. COLM, 2024. \
> [3] Yutao Sun, et al. Retentive Network: A Successor to Transformer for Large Language Models. arXiv, 2023. \
> [4] Yuhong Chou, et al. MetaLA: Unified Optimal Linear Approximation to Softmax Attention Map. NeurIPS, 2024.

---

> > ### Comment · Reviewer_xf9Q · 2025-04-01
> >
> > Thanks for the response!
> >
> > * On Q1, can you elaborate the sliding window size?
> > * On Q2, it would be great to include a variant with plain attention to empirically support the argument.
> > * On Q4, can you provide mathematic forms of the pooling operation. Sorry that it's still unclear to me the details of pooling operation.

---

> > > ### Author Response · Authors · 2025-04-03
> > >
> > > We extend our highest respect for the time and effort you devoted to reviewing our manuscript and have provided detailed responses to your further questions.
> > >
> > > > `Q1`: Can you elaborate the sliding window size?
> > >
> > > All experiment results we report on the performance of Liger are based on the sliding window size of 64 in our default implementation (see section 3.3 for details).
> > >
> > > > `Q2`: It would be great to include a variant with plain attention.
> > >
> > > Thanks for your comment. We have supplemented the result of plain linear attention variant without gating mechanism (Liger-LA) compared with Liger-GLA with gating mechanism .
> > >
> > > ||PiQA|ARC-e|ARC-c|Hella.|Wino.|MMLU|Avg.|Avg.(no MMLU)|
> > > |-|-|-|-|-|-|-|-|-|
> > > |Liger-Gated Linear Attention|80.3|81.1|52.5|76.3|72.0|43.4|67.6|72.4|
> > > |Liger-Linear Attention|79.9|79.8|52.0|75.3|70.6|38.8|66.1|71.5|
> > >
> > > We observed that in the scenario of linearization, Liger-Gated Linear Attention with gating mechanism performs better than Liger-Linear Attention (plain linear attention) without gating mechanism, which is consistent with the conclusion of related works that **gating mechanism plays an significant role in linear models**.
> > >
> > > > `Q4`: Can you provide mathematic forms of the pooling operation?
> > >
> > > The pooling operation we use is implemented using `torch.AdaptiveAvgPool1d`, and its specific mathematical form is as follows:
> > >
> > > Given an input tensor $x$ of size ($D_{in}$), the output tensor $y$ of size ($D_{out}$) is computed as follows (we ignore the batch size and sequence length here since we adopt pooling operation over $D$ dimension. Usually $D_{out} \leq D_{in}$):
> > >
> > > Let $k=\lfloor \frac{D_{in}}{D_{out}} \rfloor$ and $r = D_{in}\ \text{mod}\ D_{out}$, For each output index $i \in [0, D_{out}-1]$, compute the *start index*: $s_i =i \times k+\min(i,r)$, and the *end index*: $e_i = s_i + k $ if $i<r$ or $e_i=s_i+k-1$. The output calculation of the pooling operation is: $y[i]=\frac{1}{e_i-s_i+1} \sum\limits_{j=s_i}^{e_i}x[j]$.
> > >
> > > Once again, thank you for your meticulous review, and we are more than happy to address any further questions you may have.

---

### Decision · Program_Chairs · 2025-05-01

**Decision:**

Accept (poster)

**Comment:**

This paper presents an approach for converting pretrained softmax attention Transformers into gated linear attention Transformers, which can inherit the benefits of linear attention without having to pretrain everything from scratch. The paper outlines a series of well-motivated and empirically-tested recipe for doing this. The initial reviews raised some questions with regard to other baselines (e.g., non-gated linear attention) as well as clarity. These concerns were largely addressed in the rebuttal.

This is a well-executed and solid paper that should be of interest to the growing community of researchers working in this space, and thus I recommend that this paper be accepted.